# The standardized 12-lead fetal electrocardiogram of the healthy fetus in mid-pregnancy: A cross-sectional study

Carlijn Lempersz[1]*, Judith O. van Laar[1,2], Sally-Ann B. Clur[3], Kim M. Verdurmen[1], Guy J. Warmerdam[2], Joris van der Post[4], Nico A. Blom[3], Tammo Delhaas[5], S. Guid Oei[1,2], Rik Vullings[2]

1 Máxima Medical Centre, Department of Obstetrics and Gynaecology, Veldhoven, The Netherlands, 2 Department of Electrical Engineering, Eindhoven University of Technology, Eindhoven, The Netherlands, 3 Department of Paediatric Cardiology, Emma Children's Hospital, Amsterdam University Medical Centre, Amsterdam, The Netherlands, 4 Amsterdam University Medical Centre, Department of Obstetrics and Gynaecology, Amsterdam, The Netherlands, 5 Department of Biomedical Engineering, CARIM School for Cardiovascular Diseases, Maastricht University, Maastricht, The Netherlands

* c.lempersz@gmail.com

## Abstract

## Introduction

The examination of the fetal heart in mid-pregnancy is by ultrasound examination. The quality of the examination is highly dependent on the skill of the sonographer, fetal position and maternal body mass index. An additional tool that is less dependent on human experience and interpretation is desirable. The fetal electrocardiogram (ECG) could fulfill this purpose. We aimed to show the feasibility of recording a standardized fetal ECG in mid-pregnancy and explored its possibility to detect congenital heart disease (CHD).

## Materials and methods

Women older than 18 years of age with an uneventful pregnancy, carrying a healthy singleton fetus with a gestational age between 18 and 24 weeks were included. A fetal ECG was performed via electrodes on the maternal abdomen. After removal of interferences, a vectorcardiogram was constructed. Based on the ultrasound assessment of the fetal orientation, the vectorcardiogram was rotated to standardize for fetal orientation and converted into a 12-lead ECG. Median ECG waveforms for each lead were calculated.

## Results

328 fetal ECGs were recorded. 281 were available for analysis. The calculated median ECG waveform showed the electrical heart axis oriented to the right and inferiorly i.e. a negative QRS deflection in lead I and a positive deflection in lead aVF. The two CHD cases show ECG abnormalities when compared to the mean ECG of the healthy cohort.

**Data Availability Statement:** Data are available from the Data Governance Board of the Máxima

Medical Centre for researchers who can demonstrate that they are qualified to use confidential data. A request can be addressed to the corresponding author (carlijn_lempersz@hotmail.com) or the Data Governance Board at the Máxima Medical Centre (J. Luime, j.luime@mmc.nl).

**Funding:** This research was supported by The Dutch Technology Foundation STW (#12470), Stichting de Weijerhorst, Horizon2020 (#719500). The funders had no role in study design, data collection and analysis, decision to publish or preparation of the manuscript.

**Competing interests:** R. Vullings is a shareholder in Nemo Healthcare BV, the Netherlands. S.G. Oei initiated the scientific research from which Nemo Healthcare originated, there is no financial relationship between Nemo Healthcare and S.G. Oei. All other authors have declared that no competing interests exist.

## Discussion

We have presented a method for estimating a standardized 12-lead fetal ECG. In mid-pregnancy, the median electrical heart axis is right inferiorly oriented in healthy fetuses. Future research should focus on fetuses with congenital heart disease.

## Introduction

The fetal heart in mid-pregnancy is one of the most difficult organs to examine during the standard anomaly scan. The assessment is made difficult due to the small size of the fetal heart, its movement, and its complicated anatomy. In addition, maternal body mass index highly influences interpretability. Taking all this into consideration, a successful assessment of the heart is highly dependent on the skill of the sonographer.[1]

The timely prenatal detection of CHD has some important advantages. In the case of severe defects parents may choose to terminate the pregnancy. Where the pregnancy is continued, a prenatal diagnosis of CHD allows the parents time to prepare for the arrival of their sick child. Furthermore, it facilitates appropriate changes in obstetric and neonatal management, including intra-uterine therapy, planning of the delivery in a center with the required neonatal and cardiothoracic surgical care facilities, and timely treatment after birth. It has been demonstrated that prenatal diagnosis of CHD increases survival rates and decreases long-term morbidity. [2–6]

An additional tool for the assessment of the fetal heart in mid-pregnancy that is less dependent on human experience and interpretation is desirable. This tool could be the fetal electrocardiogram. Electrocardiography is used worldwide as a relatively simple tool to assist in the diagnosis of heart disease in adults and children as well as in the diagnosis and management of arrhythmias.

Until now, it has not been possible to record a reliable standard non-invasive fetal ECG for fetal heart assessment. Inter- and intra-fetal comparisons are hampered since the fetus is free to move underneath the transabdominal electrodes and thereby can take any orientation with respect to the electrodes. Hence, standardization of the fetal ECG, i.e. normalizing for the fetal orientation, is needed to allow fetal ECG waveform analysis. A published standardization method is currently not available. [7] Moreover, very little is known about what constitutes a normal fetal ECG at around 20 weeks of gestation.

In this paper, we present a method for standardization of the fetal ECG that was applied to a cohort of more than 300 fetuses to show the feasibility of recording a standardized fetal ECG in mid-pregnancy. Furthermore, we compared the normal ECG to the ECGs of two fetuses prenatally diagnosed with congenital heart disease (CHD) to illustrate the potential value of fetal ECG for CHD screening and diagnosis in mid-pregnancy and discuss the possible future applications of the fetal ECG.

## Materials and methods

### Ethics statement

The study was approved by the Máxima Medical Centre institutional review board (NL48535.015.14). Participants were included in the study after written informed consent had been obtained.

This study was part of a larger ongoing entity, consisting of a healthy cohort and a group of fetuses with known congenital heart disease (CHD).This trial is registered at the Netherlands Trial Register (NTR5906). The study protocol has been published by Verdurmen et al. [8] The study was approved by the Máxima Medical Centre institutional review board (NL48535.015.14). Fetal ECG measurements were performed between May 2014 and February 2017 at the Máxima Medical Centre, Veldhoven, The Netherlands and at 'Diagnostiek voor U' diagnostic center, Eindhoven, The Netherlands. Measurements were performed directly before or after the 20-week fetal anomaly scan. This anomaly examination was performed by a certified and experienced ultrasonographist. Three months after birth, the participants received a questionnaire to verify that the child was healthy and did not have CHD. This three month time interval was chosen because in The Netherlands every newborn will have several general check-ups by a primary healthcare doctor within three months after birth.

Women with an uneventful pregnancy, carrying a singleton fetus with a gestational age between 18 and 24 weeks, were included in the study after written informed consent had been obtained. The included pregnant women had to be older than 18 years of age. Fetuses with diagnosed CHD were excluded. Other exclusion criteria were multiple pregnancies, insufficient understanding of the Dutch language, and any known fetal congenital anomalies other than CHD. If the fetus turned out to have a CHD later in pregnancy or postnatally, it was subsequently excluded from further analysis.

To illustrate the potential of fetal electrocardiography for CHD screening, the normal ECG was compared to the ECG of two fetuses prenatally diagnosed with different CHDs. These ECG recordings were performed in the Amsterdam University Medical Center, Amsterdam, the Netherlands and approved by the medical ethics committee of the Amsterdam University Medical Center (2015_221#A201583).

The fetal ECG was recorded with adhesive Ag/AgCl electrodes on the abdomen of the pregnant women in a semi-recumbent position. In total, eight electrodes were placed on the abdomen in a fixed configuration (see Fig 1) in order to yield six channels of bipolar electrophysiological measurements: the other two electrodes served as common reference and ground. Application of the device is comparable to a regular ECG device and takes no more than 5 minutes. For research purposes, the duration of the registration was approximately 30 minutes, during which the fetal position was determined four times by ultrasound assessment. The determination of the fetal position typically took 10–20 seconds. After a short instruction during one measurement medical students were able to perform the measurements and the fetal orientation ultrasounds without supervision.

The electrophysiological signals were digitized and stored at 500 Hz sampling frequency by a prototype fetal monitoring system (Nemo Healthcare BV, the Netherlands) to enable analysis in a later stage. After digitization, the acquired signals were processed by PC-based signal processing techniques as previously described by Vullings et al. [9, 10] and Warmerdam et al. [11], as illustrated in Fig 2.

The first step in the process was the suppression of interferences such as the maternal ECG, powerline interference, and electromyographic signals from within the maternal body, using a template-based subtraction technique [10], a Kalman smoother [11], and a bandpass filter[10], respectively. The fetal QRS complexes were then identified using a method described in Warmerdam et al. [12] Subsequently, the fetal ECG signals were segmented in such a way that every segment contained exactly one heartbeat. The length of the segments was defined as mean RR interval (i.e. inter-beat interval) of the particular recording/fetus. The start of each segment was defined such that the R-peak was located at 40% of the segment length. As a consequence, all segments were synchronized on the position of the R-peak.

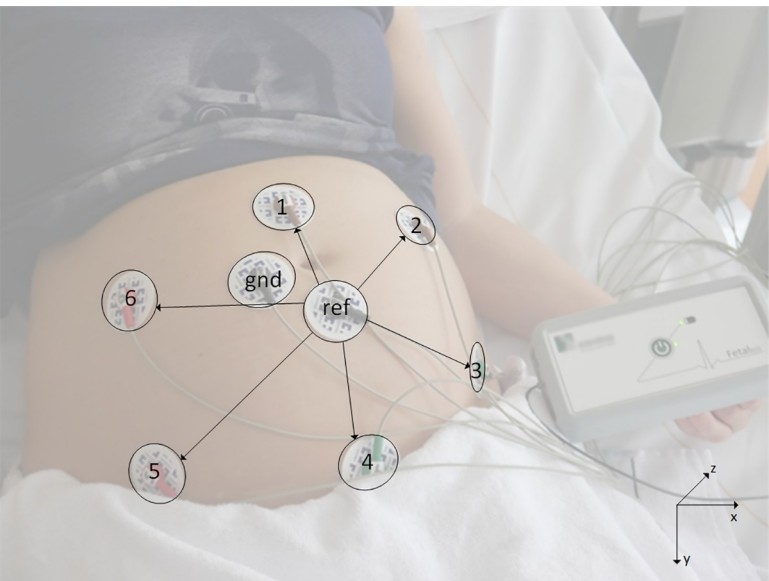

**Fig 1. Measurement setup.** Measurement setup with eight electrodes on the maternal abdomen (six recording electrodes, one common reference (ref), one ground (gnd)) and the prototype fetal monitoring system. The bipolar channels are indicated by the arrows and formed by the electrodes 1–6 with respect to the common reference (eg 1 – ref, 2 –ref). The positions of the electrodes and lead vectors of the recorded channels are defined within the xyz-axis system depicted on the bottom right.

The QRS detection method of Warmerdam et al. [12] was used to check whether a detected fetal QRS complex was correct or not. This system uses various checks: the interval between successive QRS complexes (i.e. RR-interval) should be in line with the physiological range of

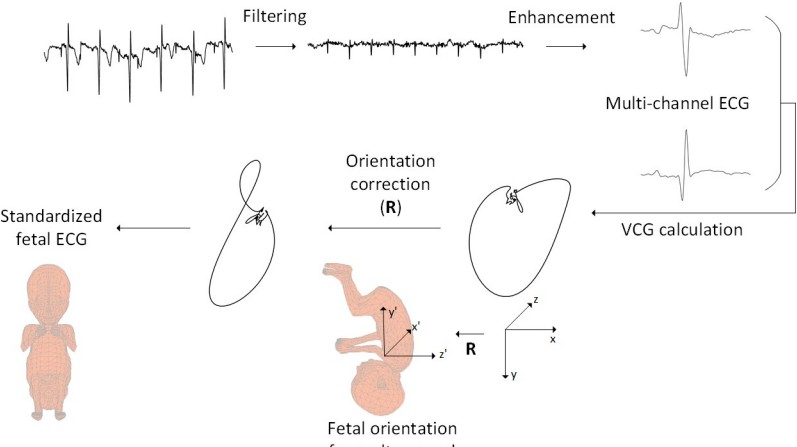

**Fig 2. Schematic illustration of signal processing to obtain standardized fetal ECG.** Schematic illustration of signal processing steps followed to obtain the standardized fetal ECG. From the top left, the consecutive steps are depicted in a clockwise direction. In the first step the raw data was filtered to suppress the maternal ECG. In the next step, the fetal ECG was further enhanced by averaging the ECG over 30 heartbeats. Multi-channel ECG complexes were subsequently combined to calculate the vectorcardiogram (VCG). This VCG is described within a coordinate system (xyz) that is defined with respect to the maternal body. Based on the fetal orientation assessed by ultrasound examination, a mathematical rotation **R** of the VCG was performed to convert the VCG to a coordinate system (x'y'z') that is defined with respect to the fetal body. Finally, the rotated VCG was transformed with the Dower matrix to yield an estimate for the standardized, 12-lead fetal ECG.

the fetal heartbeat (i.e. between 60 and 240 beats-per-minute) and cannot vary more than 20% between consecutive RR-intervals. Moreover, the morphology of the QRS complexes should be similar between consecutive heartbeats, their energy not be higher than physiologically plausible, and they should not coincide for more than three consecutive heartbeats with the ECG of the mother. The latter criterion was used to prevent the erroneous detection of maternal ECG residues as fetal QRS complexes. When a detected fetal QRS complex did not confirm to all these criteria, it was rejected and excluded from further analysis. We used an average of at least 25 detected fetal QRS complexes per minute as threshold for signal quality. Where this threshold was not met, the entire recording was excluded from further analysis due to low signal quality.

Where the signal quality was assessed as adequate, we enhanced the fetal ECG further by averaging 30 consecutive segments. This procedure was performed for each of the six channels of fetal ECG data.

As mentioned previously, before we could compare ECG waveforms between patients, the ECG needed to be normalized for the fetal orientation. Without such normalization, a specific electrode would record a different ECG waveform for, for example, a fetus in a cephalic position versus the same fetus in a breech position. To normalize for fetal orientation, we first calculated the vectorcardiogram (VCG) of the fetus. [9] This VCG entailed a three-dimensional representation of the fetal electrical cardiac activity, in other words the path of electrical cardiac activation through the heart. The VCG was calculated for each average ECG (i.e. the ECG obtained from averaging 30 consecutive segments). At this point, the fetal VCG was determined in the xyz-coordinate system that was described with respect to the maternal body (see Fig 1, bottom right). In order to facilitate interpretation and standardization of this fetal VCG, it must be placed within a coordinate system that is described with respect to the fetal body. The mathematical rotation that defines this conversion between coordinate systems could be calculated based on an ultrasound assessment of the fetal orientation.

Furthermore, the expectation-maximization algorithm was used to track rotations of the VCG between heartbeats due to fetal movements between the ultrasound assessments (Fig 3).

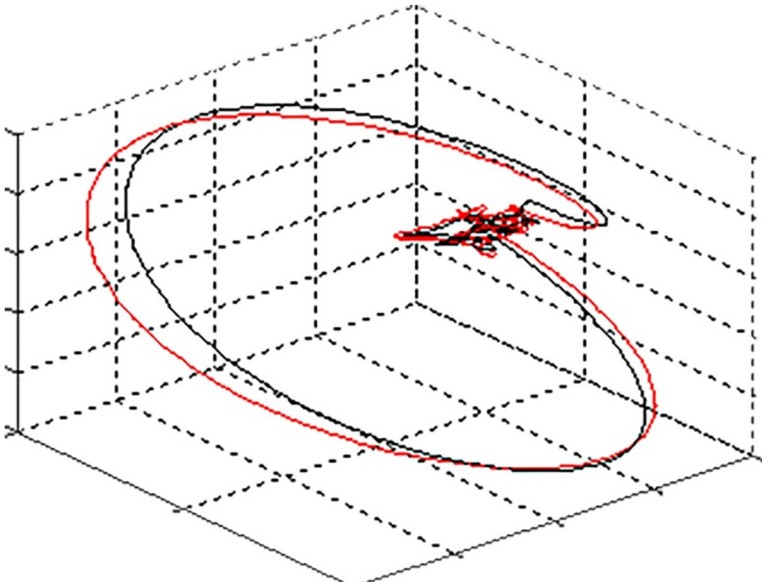

**Fig 3. Rotation of the VCG between heartbeats due to fetal movement.** In black the VCG at time t = 0, in red the VCG at time t = 1. Due to fetal movement there was a rotation of the VCG.

[13] Based on the tracked VCG rotations, we corrected for these fetal movements, using similar mathematical rotations as used for the correction of the fetal orientation. The four ultrasound assessments during the measurements were used to correct for cumulative errors in this movement correction method and to determine the initial orientation of the fetus. After correcting for fetal movements, the VCGs throughout the entire recording were averaged to further enhance signal quality.

Because clinicians are not used to interpreting a VCG, we chose to visualize the fetal cardiac activity by means of a 12-lead ECG. As described by Dower [14], in adult electrocardiography the VCG can be used to calculate standardized ECG leads. In this study, we used the Dower transformation [15] to calculate a 12-lead ECG from the VCG for each fetus.

The segmentation of the fetal ECG data described above depended on the fetal heart rate during the recording: the length of the segments was defined as the mean RR interval of the fetus during the recording. Because different fetuses have different heart rates, the calculated standardized ECG leads had different lengths across the fetuses. To enable comparison between fetuses, we calculated the average RR interval over all fetuses: in this case a length of 0.44s. Next, we resampled all standardized ECG leads to match this length. As, the R-peak was located at 40% of the segment length for each fetus, after resampling to the uniform segment length of 0.44s, the R-peaks for all fetuses were still at 40% and hence synchronized over all standardized ECG leads.

To assess whether a standardized fetal ECG is feasible, we determined the median amplitude and interquartile range (IQR) for all fetuses with adequate signal quality to yield a median ECG waveform.

## Results

### Normal hearts

During the study period, informed consent was obtained from 328 participants and a fetal ECG was performed. 37 measurements were excluded from further analysis because we did not receive a postpartum questionnaire (n = 14) or because there was no fetal orientation ultrasound available (n = 23). In addition two children appeared to have a CHD and three children were diagnosed with a syndrome and were excluded from further analysis. Of the final 286 recordings, 281 were of adequate quality to generate a standardized fetal ECG, yielding a success rate of 98.3%.

The number of detected QRS complexes in the recordings with sufficient quality ranged from 35 to 156 complexes per minute and from 565 to 6130 detected complexes over the full recording.

Fig 4 shows the median ECG waveform for leads I (left) and aVF (right) in black, with in grey the IQR for the included 299 fetuses.

In Fig 5 the median 12-lead ECG is shown.

From all the figures, it can be seen that the median ECG waveform indicates an electrical heart axis oriented right inferiorly, based on the negative QRS deflection in lead I and the positive deflection in lead aVF. This suggests a dominant right ventricle, which is in line with the higher cardiac output (volume loading) and pressure loading of this ventricle in-utero. [16–19]

### Congenital heart disease

**ECGs were recorded from two fetuses diagnosed with CHD.** The first patient was diagnosed at 21+2 weeks of gestation with left atrial isomerism (mesocardia, bilateral superior caval veins and polysplenia) and complete atrio-ventricular septal defect (AVSD) with left

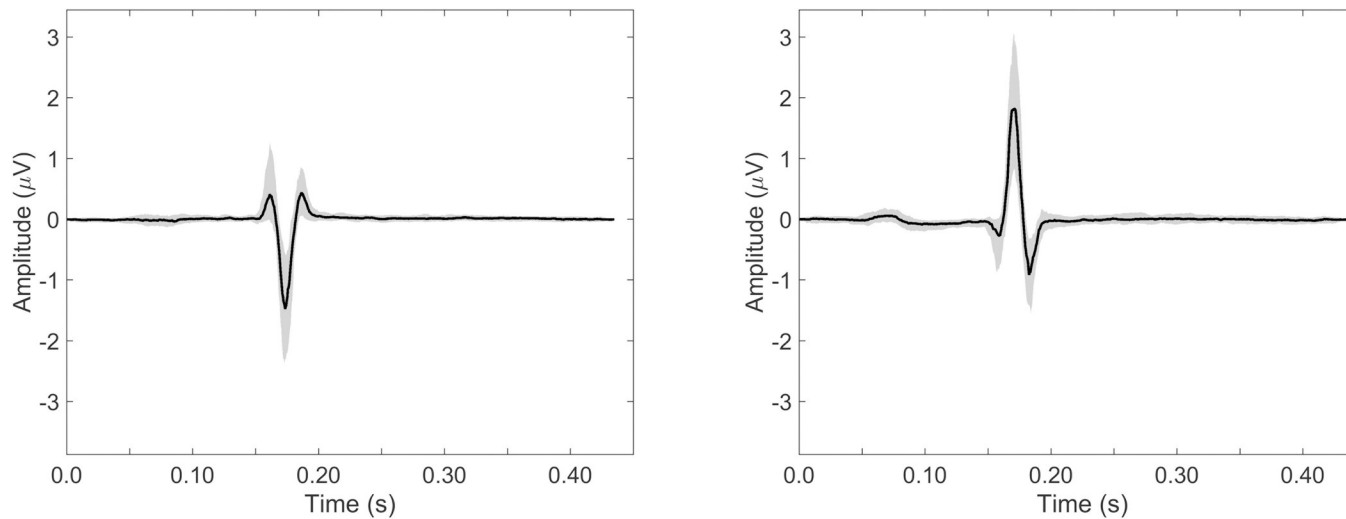

**Fig 4. Normal ECG waveform.** Normal ECG waveform with in black the median over 281 healthy fetuses and in grey the interquartile range, shown for lead I (A) and lead aVF (B).

ventricular dominance. The fetal ECG registration was made at 26+3 weeks of gestation. Postnatally, this baby died suddenly from a group B streptococcal septicemia and the cardiac diagnosis was confirmed at post mortem examination.

The second patient was diagnosed at 16+4 weeks of gestation with a hypoplastic right ventricle, tricuspid stenosis and a dysplastic pulmonary valve (hypoplastic right heart). The fetal ECG registration was made at 20+6 weeks of gestation. The diagnosis was confirmed by post mortem examination after termination of the pregnancy.

In Fig 6, the ECG of the fetus with left atrial isomerism and AVSD can be seen. The QRS axis is abnormal (-45 degrees) and there is a prominent left ventricle. In Fig 7, the ECG of the fetus with the hypoplastic right heart (a ductal dependent lesion) is depicted. The QRS axis is abnormal (+60 degrees) and there is left ventricular dominance, based on the prominent R waves in V5 and V6.

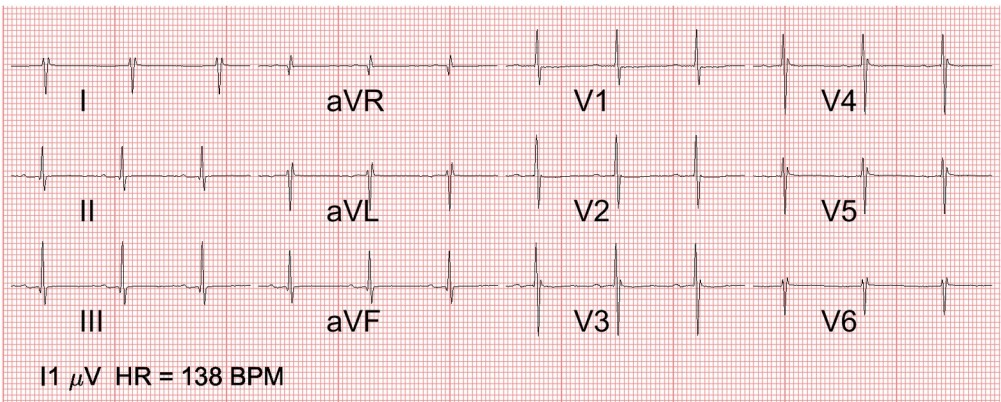

**Fig 5. 12-lead normal fetal ECG in mid-pregnancy.** 12-lead ECG of the normal fetal heart at 20 weeks of gestation, calculated as the median over 281 healthy fetuses. Note the rightward QRS axis and the right ventricular dominance (positive R wave in V1 with deep S in V6 and failure of R wave progression precordially). The marker on the bottom left indicates the scale at which an amplitude of 1 μV is depicted.

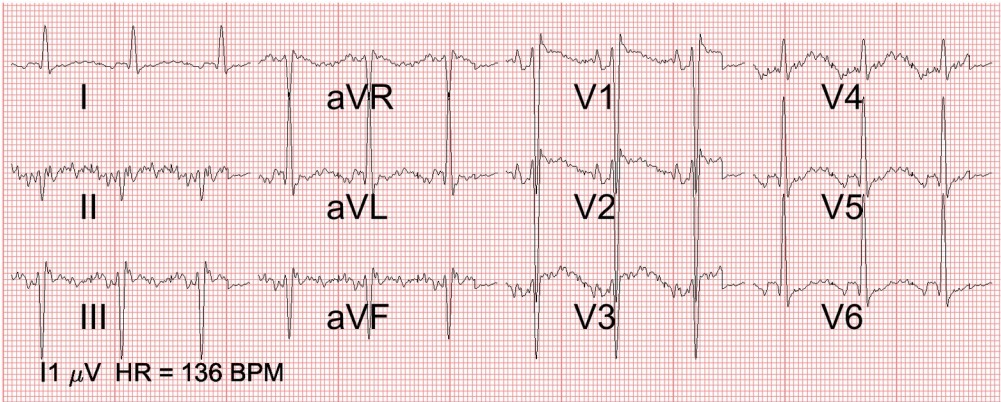

**Fig 6. 12-lead fetal ECG of left isomerism and atrioventricular septal defect.** 12-lead ECG of case 1 with the atrioventricular septal defect: Note the abnormal QRS axis (-45 degrees) and prominent left ventricle (prominent R in V5 and V6 and deep S in V1 and V2 as well as the negative aVR). The marker on the bottom left indicates the scale at which an amplitude of 1 μV is depicted.

To illustrate the difference between these two cases and the normal fetal ECG, in Fig 8 the ECGs of the CHD cases are overlayed on the IQR of the normal for leads I and aVF.

## Discussion

For the first time, we have presented the feasibility of a method for estimating a standardized 12-lead ECG for a healthy fetus around 20 weeks of gestation in which we used ultrasound assessment of the fetal orientation to correct for its influence on the estimated fetal ECG.

When compared to other organs examined during the 20-week anomaly scan, screening of the fetal heart with ultrasound imaging is regarded as the most difficult due to its motion, small size and anatomical complexity. Therefore, CHD in the mid-term fetus is often missed. [20, 21] This is especially undesirable because it has been demonstrated that prenatal diagnosis of CHD increases neonatal survival rates and decreases neonatal long-term morbidity. [4–6,

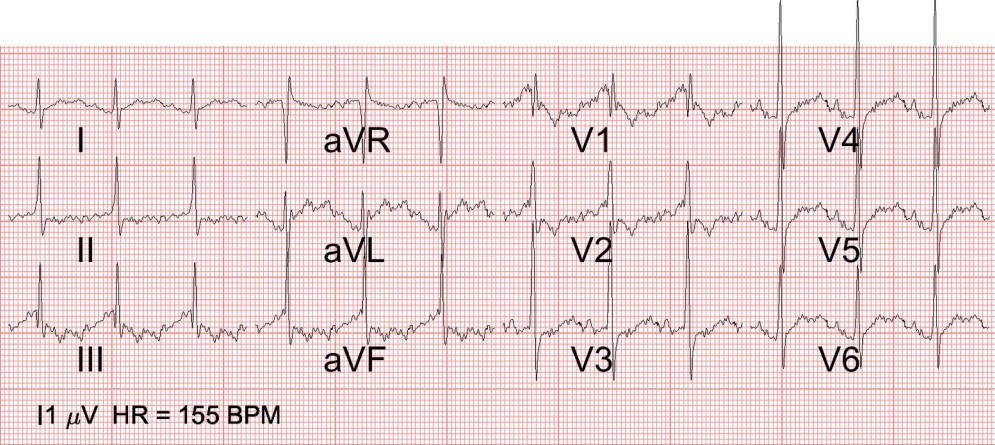

**Fig 7. 12-lead fetal ECG of hypoplastic right heart.** 12-lead ECG of case 2 with a hypoplastic right heart (hypoplastic right ventricle, tricuspid stenosis and dysplastic pulmonary valve): Note the abnormal QRS axis (+60 degrees) and left ventricular dominance (prominent R waves in V5 and V6). The marker on the bottom left indicates the scale at which an amplitude of 1 μV is depicted.

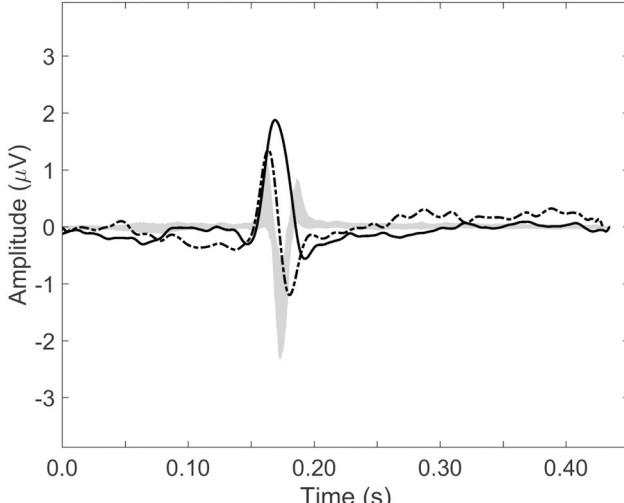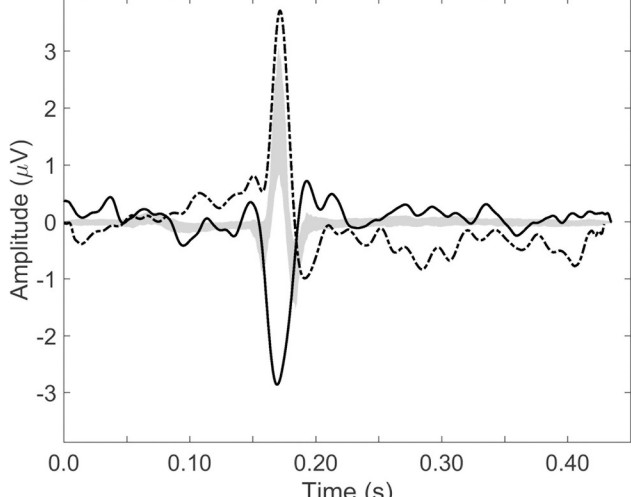

**Fig 8. ECG of cases compared to normal ECG.** Leads I (A) and aVF (B) for the two CHD cases, plotted together with the IQR of the normal ECG. The gray area represents the IQR of the normal ECG, the solid line represents the fetus with atrio-ventricular septal defect (Fig 6) and the dashed line represents the fetus with hypoplastic right heart (Fig 7).

22, 23] The performance of a fetal ECG could aid screening for CHD in mid-pregnancy in primary care and follow-up in dedicated centers, since it is independent of the experience of the sonographer, difficult fetal imaging due to maternal obesity, an unfavorable fetal position, or reduced amniotic fluid. The fetal ECG might also give more information about the evolution of the CHD during pregnancy and, with that, the health status of the fetus. Although there is no reference to compare our results with, our results show the reproducibility of the 'mean' ECG in all 281 participating fetuses. Furthermore, it can be seen that the standardized ECG has, conform expectations, a right ventricular dominance e.g. a right oriented electrical heart axis. The electrical heart axis represents the median vector of the electrical activity through the heart during one cardiac cycle and gives information about the muscle distribution of the heart. CHD may alter this distribution. The electrical heart axis could thus be a potential indicator for CHD in mid-pregnancy.

The potential use of the fetal ECG in CHD screening was further illustrated by the examination of ECG recordings from the two cases with CHD (Figs 6 and 7). Both cases showed ECG abnormalities (e.g. abnormal QRS axis) when compared to the IQR of the normal ECG (Fig 8).

In the first CHD case, left atrial isomerism with AVSD, there was an abnormal axis due to the altered cardiac conduction system anatomy (causing a left anterior hemi-block) and left ventricular hypertrophy due to the atrio-ventricular valve regurgitation in utero. This case died unexpectedly of septicemia as a neonate but had this not occurred, we would have expected the baby to develop cardiac failure from around two weeks of age and she would have required a complete surgical correction before the age of three months. Case two was a ductal dependent lesion where a prenatal detection is extremely important to ensure the timely postnatal administration of intravenous prostaglandins to keep the arterial duct open. Failure to do so could be life-threatening due to inadequate pulmonary perfusion resulting in acidosis, organ failure, neurological damage and death. Both cases could have been detected by fetal ECG screening.

The high success rate of 98.3% shows the promise of a reliable additional tool for clinical practice, which may be less subject to human interpretation and experience. Besides signal quality, the applicability of this method also depends on the availability to estimate the fetal orientation (e.g. from simultaneous ultrasound examination). Without accurate information

on the fetal orientation, normalization for this orientation is not possible and analysis of the fetal ECG has to be limited to the analysis of ECG intervals.

## Limitations

The signal analysis methods that were used in our study to enable standardization of the fetal ECG have four main limitations. First, in the calculation of the VCG, information of the amplitude of the fetal ECG and VCG was lost, because the distance between the fetal heart and each of the transabdominal electrodes is different. The thickness of the layers of maternal tissue in between the fetal heart and the electrode varies between the different electrodes and therefore, attenuation of the ECG signal due to conduction of the signal from the heart to electrode will be different for each electrode. Our method for VCG calculation can compensate for such variations in attenuation, but only with normalization of all amplitudes. Besides compensating for inter-electrode differences in ECG signal attenuation, this method has the capacity that inter-patient variations in ECG signal attenuation are also compensated for. Such inter-patient variations could originate from differences in BMI, amount of amniotic fluid, distribution of muscle and fat tissue in the abdomen, and properties of the skin. Second, the calculation of the 12-lead ECG from the VCG via the Dower matrix is based on assumptions about geometrical and conductive properties of the adult thorax that may not fully apply to the fetal thorax. Interpretation of the fetal ECG should therefore not be based on guidelines used for 12-lead adult ECG. The third limitation is that to further enhance signal quality, averaging over 30 segments (i.e. heartbeats) took place. This entails a trade-off between gain in fetal ECG signal quality and loss of inter-beat variability in the ECG. The gain in signal quality allows for possible diagnosis of structural malformations of the heart, but the concomitant loss of inter-beat variability hampers arrhythmia diagnosis.

Averaging over multiple heartbeats emphasizes the part of the ECG that is common between heartbeats. Structural malformations will affect every heartbeat in more or less the same way and hence be visible in the average ECG. The fourth limitation is that the data was evaluated retrospectively. However, pseudo real-time implementations (i.e. only a few seconds delay) of the described technology are being developed.

Future research should focus on defining normal ranges and values for the fetal ECG in mid-pregnancy, including the electrical heart axis. When normal ranges and values are established research could focus on the application of the fetal ECG for the detection and follow-up of fetal heart disease. An abnormal fetal ECG may expedite a referral for an advanced fetal echocardiogram in dedicated centers in the future.

## Conclusion

To conclude, we demonstrated that it is possible to determine a fetal ECG for a healthy fetus at 20-weeks of gestation and standardize this ECG for the fetal orientation. As a result, we have presented the first standardized ECG for a healthy 20-week fetus.

To illustrate the clinical relevance of this standardized fetal ECG, we showed that the standardized ECG thus derived is clearly different from 2 cases with congenital heart disease. Although the recording of a 12-lead fetal ECG is feasible with non-invasive fetal ECG technology, more research is needed to study its implications for clinical practice.

## Supporting information

**S1 Checklist.**
(PDF)

**S1 Data.**
(PDF)

## Acknowledgments

The authors would like to express their gratitude to Drs. Aimee van Dobben-Rodenburg from 'Diagnostiek voor U' and to N. Eijsvoogel, D. Aben, M. Sengers, J. Drinkwaard, C. van den Oord, M. van Wierst, O. Hulsenboom, L. Noben, L. Cornelissen for their role in the data collection.

## Author Contributions

**Conceptualization:** Carlijn Lempersz, Judith O. van Laar, S. Guid Oei, Rik Vullings.

**Data curation:** Carlijn Lempersz, Judith O. van Laar, S. Guid Oei.

**Formal analysis:** Rik Vullings.

**Funding acquisition:** S. Guid Oei, Rik Vullings.

**Investigation:** Carlijn Lempersz, Judith O. van Laar, Sally-Ann B. Clur, Kim M. Verdurmen, Guy J. Warmerdam.

**Methodology:** Carlijn Lempersz, Kim M. Verdurmen, Guy J. Warmerdam, Tammo Delhaas, S. Guid Oei, Rik Vullings.

**Project administration:** Carlijn Lempersz, Judith O. van Laar, Sally-Ann B. Clur, Joris van der Post, Nico A. Blom, Tammo Delhaas, Rik Vullings.

**Software:** Guy J. Warmerdam, Rik Vullings.

**Supervision:** Judith O. van Laar, S. Guid Oei, Rik Vullings.

**Writing – original draft:** Carlijn Lempersz, Rik Vullings.

**Writing – review & editing:** Judith O. van Laar, Sally-Ann B. Clur, Kim M. Verdurmen, Guy J. Warmerdam, Joris van der Post, Nico A. Blom, Tammo Delhaas, S. Guid Oei.

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
