## [Decision Letter · Decision Letter 0]

24 Jan 2020

PONE-D-19-28704

The Standardized 12-lead Fetal Electrocardiogram of the Healthy Fetus in Mid-Pregnancy: a cross-sectional study

PLOS ONE

Dear ms Lempersz,

Thank you for submitting your manuscript to PLOS ONE. After careful consideration, we feel that it has merit but does not fully meet PLOS ONE’s publication criteria as it currently stands. Therefore, we invite you to submit a revised version of the manuscript that addresses the points raised during the review process.

We would appreciate receiving your revised manuscript by Mar 09 2020 11:59PM. To enhance the reproducibility of your results, we recommend that if applicable you deposit your laboratory protocols in protocols.io, where a protocol can be assigned its own identifier (DOI) such that it can be cited independently in the future. For instructions see: http://journals.plos.org/plosone/s/submission-guidelines#loc-laboratory-protocols

We look forward to receiving your revised manuscript.

Kind regards,

Leticia Reyes

Academic Editor

PLOS ONE

Journal Requirements:

 [No statement provided]

Please provide an amended Funding Statement that declares *all* the funding or sources of support received during this specific study (whether external or internal to your organization) as detailed online in our guide for authors at http://journals.plos.org/plosone/s/submit-now Please state what role the funders took in the study.  If any authors received a salary from any of your funders, please state which authors and which funder. If the funders had no role, please state: "The funders had no role in study design, data collection and analysis, decision to publish, or preparation of the manuscript."

 [No statement provided]

a. Please complete your Competing Interests statement to state any Competing Interests.

If you have no competing interests, please state "The authors have declared that no competing interests exist.", as detailed online in our guide for authors at http://journals.plos.org/plosone/s/submit-now

5. Please include captions for your Supporting Information files at the end of your manuscript, and update any in-text citations to match accordingly. Please see our Supporting Information guidelines for more information: http://journals.plos.org/plosone/s/supporting-information

Reviewers' comments:

Reviewer's Responses to Questions

**Comments to the Author**

1. Is the manuscript technically sound, and do the data support the conclusions?

Reviewer #1: Yes

Reviewer #2: Partly

2. Has the statistical analysis been performed appropriately and rigorously? 

Reviewer #1: Yes

Reviewer #2: I Don't Know

3. Have the authors made all data underlying the findings in their manuscript fully available?

Reviewer #1: Yes

Reviewer #2: Yes

4. Is the manuscript presented in an intelligible fashion and written in standard English?

Reviewer #1: Yes

Reviewer #2: No

5. Review Comments to the Author

Reviewer #1: The study is well-designed and the paper is well-written. The feasibility to obtain fetal non-invasive ECG before 26 weeks of gestation is known (lack of vernix caseosa). The manuscript contains a piece of novel information on fetal non-invasive ECG as a screening method for fetal cardiac malformations. The authors have used vector ECG as a main diagnostic tool. The results are promising. I have not found any principal shortcomings. My recommendation is to accept. I have several questions.

1. What is the best term for fetal non-invasive ECG screening?

2. Have you found fetal non-invasive ECG intervals and peaks measurement feasible for clinical interpretation?

3. It's known that the predominance of right ventricle activity is typical during fetal life in a healthy pregnancy. Where is the border between health and pathology in her projections on fetal vector ECG?

Reviewer #2: Overall the research idea and number of recordings collected is significant, however, there are several factors that greatly decrease the quality of the article. The structure is quite confusing and the reader sometimes feels lost. In some parts (chapter Materials and Methods) the authors supply the reader with a lot of details that are not necessary (or could be summarized using e.g. a Table) whereas some important information is missing.

Line 216 and 304 – You state that the success rate was 98.87 %. What did you use for the comparison? There is no reference mentioned. Furthermore, you state that “there is no reference to compare our results with” (line 281).

Line 217-218 – you mentioned that the number of detected QRS complexes in the recordings with sufficient quality ranged from 35 to 156 complexes per minute and from 565 to 6130 detected complexes over the full recording. Again, is there any reference annotation by the experts? Or how do you differentiate between the fetal QRS complexes in the accurately extracted signal from e.g. the maternal residue in the signals extracted poorly?

The above mentioned is either not defined at all or it is described very poorly (e.g. lines 146-157) so it is very hard to follow the research process. I guess the flowchart in the last figure was created to help in this matter, but I personally find it more misleading than helpful.

What I am really confused from the most is the Conclusion. It includes only 3 sentences and states something else that what my idea about the article was. In Results you mention that you recorded 328 fECGs and analyzed 281 from it. However, in the Conclusion it seems like you presented a single ECG for 20-week fetus and then compared a mean ECG thus derived with 2 cases with congenital heart disease. So this article a case study??

6. PLOS authors have the option to publish the peer review history of their article (what does this mean?). If published, this will include your full peer review and any attached files.

Reviewer #1: Yes: Igor V. Lakhno

Reviewer #2: No

---

## [Author Response · Author response to Decision Letter 0]

12 Mar 2020

Dear editor and reviewers,

First we would like to thank you and the journal for considering our work for publication in PLOS ONE. We have carefully considered the reviewers’ remarks and adjusted the manuscript accordingly. For our response to the comments we would like to refer to the text below. For your and the reviewers’ convenience, any changes to the text that were made in response to the reviewers’ comments are highlighted in the redline version of the manuscript.

Reviewer #1.

The study is well-designed and the paper is well-written. The feasibility to obtain fetal non-invasive ECG before 26 weeks of gestation is known (lack of vernix caseosa). The manuscript contains a piece of novel information on fetal non-invasive ECG as a screening method for fetal cardiac malformations. The authors have used vector ECG as a main diagnostic tool. The results are promising. I have not found any principal shortcomings. My recommendation is to accept. 

Thank you for your positive evaluation of our manuscript.

I have several questions. 

1. What is the best term for fetal non-invasive ECG screening?

The best term is between 20-22 weeks of gestation. In this study, we found good signal quality from the 20th week of gestation. Furthermore, when screening for congenital heart disease it is important to screen as early as possible. This gives parents the chance to make an unrushed decision as to whether to continue with the pregnancy or not should severe congenital heart disease be detected. Where the pregnancy is continued, preparations in dedicated centres can be made to give optimal care for mother and child. Changes in the introduction have been made where this is addressed.

2. Have you found fetal non-invasive ECG intervals and peaks measurement feasible for clinical interpretation?

Due to averaging of the fetal ECG over multiple heartbeats we found that some information regarding possible variations in ECG intervals is lost. In the averaged ECG complexes, QRS complexes and often also P-waves can be identified; T-waves appear to be more challenging, probably due to the effect of the tissues between the fetal heart and the abdominal electrodes. These tissues are known to suppress low-frequency waves, such as the T-wave. More research regarding interval analysis is needed and is a subject we hope to address in the near future. 

We think, that analysis of the peak amplitudes is possible, under the constraint that we have to compare amplitudes relative to each other, for instance T/QRS ratios. Due to inter- and intra-patient variations in the position of the fetus and its proximity to the abdominal electrodes, the amplitude of the recorded ECG might vary, complicating inter-patient comparisons in absolute peak amplitudes. This is now better explained in the Discussion section of the manuscript. 

3. It’s known that the predominance of right ventricle activity is typical during fetal life in a healthy pregnancy. Where is the border between health and pathology in her projections on fetal vector ECG?

This border is not defined yet. We are working with this dataset to define normal ranges for the electrical heart axis in healthy fetuses in mid-pregnancy. Then we will compare those results with the electrical heart axis found in fetuses with a known congenital heart disease to indeed search for such border. The answer to this question is added in the Discussion section of the manuscript. 

Reviewer #2

Overall the research idea and number of recordings collected is significant, however, there are several factors that greatly decrease the quality of the article. The structure is quite confusing and the reader sometimes feels lost. In some parts (chapter Materials and Methods) the authors supply the reader with a lot of details that are not necessary (or could be summarized using e.g. a Table) whereas some important information is missing.

Thank you for your feedback and questions to help us clarify the study and its results better. We have changed the methods (see marked changes) to try make them more clear.

1. Line 216 and 304 – You state that the success rate was 98.87 %. What did you use for the comparison? There is no reference mentioned. Furthermore, you state that “there is no reference to compare our results with” (line 281).

The success rate is defined as the percentage of recordings with sufficient quality for further analysis. The definition of “sufficient quality” is defined in the manuscript. Indeed, in antepartum fetal ECG measurements, there is no reference measurement to validate the detected fetal ECG. Hence, we used quality criteria that include the interval between detected fetal QRS complexes (should be comparable between beats) and their morphology (amplitude should not be too large to be physiological). This is stated in the methods section. From the 286 recordings used in our study, 281 were of sufficient quality for further analysis. This is 98.3% of the recordings. The definition of quality has been described more clearly in the Materials and Methods section of the manuscript.

2. Line 217-218 – you mentioned that the number of detected QRS complexes in the recordings with sufficient quality ranged from 35 to 156 complexes per minute and from 565 to 6130 detected complexes over the full recording. Again, is there any reference annotation by the experts? Or how do you differentiate between the fetal QRS complexes in the accurately extracted signal from e.g. the maternal residue in the signals extracted poorly? The above mentioned is either not defined at all or it is described very poorly (e.g. lines 146-157) so it is very hard to follow the research process. I guess the flowchart in the last figure was created to help in this matter, but I personally find it more misleading than helpful.

Because we included almost 300 recordings of about 20 minutes each in our dataset, the number of fetal QRS complexes probably exceeded 750.000 which is too laborious to validate by expert referees and as such prone to error. We used criteria mentioned in our reply to the comment above to verify whether a detected QRS complex is indeed a QRS complex. This methodology has been validated and described in more detail in Warmerdam et al. (1). These criteria also include that the detected fetal QRS complexes should not coincide for more than a few beats with the detected maternal QRS complexes. If this were to happen, indeed the detected fetal QRS complexes might be a residue of the maternal ECG and should be excluded from further analysis. 

We have adapted the manuscript to describe our quality assessment in more detail. 

3. What I am really confused from the most is the Conclusion. It includes only 3 sentences and states something else that what my idea about the article was. In Results you mention that you recorded 328 fECGs and analyzed 281 from it. However, in the Conclusion it seems like you presented a single ECG for 20-week fetus and then compared a mean ECG thus derived with 2 cases with congenital heart disease. So this article a case study??

As stated in the introduction, our main goal was to investigate if it is possible to produce a standardized fetal ECG. For the 281 fetuses included in our analysis (i.e. those with sufficient signal quality) we analysed whether there it is indeed possible to standardize the ECG measurement by correcting for the fetal orientation. Based on our results, we can conclude that the fetal ECGs, after correction for orientation, look very similar and that the inter-patient correspondence is equivalent to that of healthy newborn ECGs.

To illustrate the potential of this technology for e.g. differentiating between healthy fetuses and those with congenital heart disease (CHD), we presented two cases of CHD to show that indeed these seem very different from the standardized ECG of a healthy fetus. This standardized ECG was determined as the median over all healthy fetuses, similar as one would describe the standard ECG in a healthy adult.

To describe our primary goal better, we have modified the conclusion.

---

## [Decision Letter · Decision Letter 1]

20 Apr 2020

The Standardized 12-lead Fetal Electrocardiogram of the Healthy Fetus in Mid-Pregnancy: a cross-sectional study

PONE-D-19-28704R1

Dear Dr. Lempersz,

We are pleased to inform you that your manuscript has been judged scientifically suitable for publication and will be formally accepted for publication once it complies with all outstanding technical requirements.

With kind regards,

Leticia Reyes

Academic Editor

PLOS ONE

Additional Editor Comments (optional):

Reviewers' comments:

Reviewer's Responses to Questions

**Comments to the Author**

1. If the authors have adequately addressed your comments raised in a previous round of review and you feel that this manuscript is now acceptable for publication, you may indicate that here to bypass the “Comments to the Author” section, enter your conflict of interest statement in the “Confidential to Editor” section, and submit your "Accept" recommendation.

Reviewer #1: All comments have been addressed

Reviewer #2: All comments have been addressed

2. Is the manuscript technically sound, and do the data support the conclusions?

Reviewer #1: Yes

Reviewer #2: Yes

3. Has the statistical analysis been performed appropriately and rigorously? 

Reviewer #1: Yes

Reviewer #2: Yes

4. Have the authors made all data underlying the findings in their manuscript fully available?

Reviewer #1: Yes

Reviewer #2: No

5. Is the manuscript presented in an intelligible fashion and written in standard English?

Reviewer #1: Yes

Reviewer #2: (No Response)

6. Review Comments to the Author

Reviewer #1: I think that your paper is focused on a very considerable issue in perinatal medicine. You have found a novel approach to the diagnosing of fetal cardiac malformations. The results look sound.

Reviewer #2: Dear authors,

Thank you for addressing my comments, great work! My recommendation is to Accept. Good luck in your future research!

7. PLOS authors have the option to publish the peer review history of their article (what does this mean?). If published, this will include your full peer review and any attached files.

Reviewer #1: Yes: Igor V. Lakhno

Reviewer #2: Yes: Ing. Radana Kahankova, PhD

---

## [Editor Report · Acceptance letter]

22 Apr 2020

PONE-D-19-28704R1 

The Standardized 12-lead Fetal Electrocardiogram of the Healthy Fetus in Mid-Pregnancy: a cross-sectional study 

Dear Dr. Lempersz:

I am pleased to inform you that your manuscript has been deemed suitable for publication in PLOS ONE. Congratulations! Your manuscript is now with our production department. 

With kind regards,

on behalf of

Dr. Leticia Reyes 

Academic Editor

PLOS ONE